# Clinical and Molecular Basis of Hepatocellular Carcinoma after Hepatitis C Virus Eradication

**DOI:** 10.3390/pathogens11040430

**Published:** 2022-04-01

**Authors:** Natsumi Oe, Haruhiko Takeda, Yuji Eso, Atsushi Takai, Hiroyuki Marusawa

**Affiliations:** 1Department of Gastroenterology, Red Cross Osaka Hospital, Osaka 5438555, Japan; kuromame723nimame@gmail.com; 2Department of Gastroenterology and Hepatology, Graduate School of Medicine, Kyoto University, Kyoto 6068501, Japan; htakeda@kuhp.kyoto-u.ac.jp (H.T.); yujieso@kuhp.kyoto-u.ac.jp (Y.E.); atsushit@kuhp.kyoto-u.ac.jp (A.T.)

**Keywords:** hepatitis C virus, liver cancer, SVR

## Abstract

Hepatocellular carcinoma (HCC) arises in the background of chronic liver diseases, including hepatitis and liver cirrhosis caused by hepatitis C virus (HCV) infection. It is well known that HCV eradication using antiviral drugs can efficiently inhibit hepatocarcinogenesis. Recent advances in and development of direct-acting antiviral (DAA) drugs has revolutionized the treatment of HCV infection, and the vast majority of HCV patients can achieve HCV eradication using DAAs. However, mounting evidence clearly indicates that HCC inevitably occurs in a subset of patients after successful viral eradication using DAA therapy. Cancer is a genetic disease, and the accumulation of genetic and epigenetic aberrations may cause hepatocarcinogenesis in chronically damaged liver, even after virus elimination. In this review, we highlight HCC development after HCV eradication and discuss the current understanding of the molecular mechanisms of tumorigenesis after virus elimination, focusing on the genetic and epigenetic background of chronically damaged liver tissues.

## 1. Introduction

Hepatocellular carcinoma (HCC) is the sixth most prevalent and fourth most lethal cancer worldwide [1]. HCC development is associated with various risk factors, including infection with hepatitis B or hepatitis C virus (HCV), obesity, alcohol consumption, hepatic steatosis, and exposure to aflatoxin B1. HCV infection is a leading cause of hepatocarcinogenesis, especially in North America, Europe, and East Asia [2]. About 70% of people with HCV develop chronic hepatitis, and 15–30% of HCV patients progress to cirrhosis within 20 years. The reported annual incidence of HCC in chronic hepatitis patients infected with HCV is 1–4%, but rises to 3–8% in cases of HCV-related liver cirrhosis [3], which is higher than that in patients with non-alcoholic steatohepatitis (NASH)-related cirrhosis (1–2%) [4]. Given recent advances in antiviral treatment, especially direct-acting antivirals (DAA), most HCV patients achieve the persistent eradication of the viruses, the so-called sustained viral response (SVR). It is widely recognized that HCV eradication can markedly reduce the incidence of HCC. However, it should be noted that HCC may inevitably recur in a subset of patients after achievement of SVR [5,6]. In this review, we highlight the clinical and basic aspects of hepatocarcinogenesis after HCV eradication and discuss recent findings on the molecular basis of post-SVR HCC development.

## 2. Progress of Anti-HCV Therapy and Its Protective Effect against Hepatocarcinogenesis

Antiviral therapy against HCV has significantly advanced over the past decade. Interferon (IFN)-α was the first drug to be approved as an antiviral therapy for patients with HCV infection [7]. However, the SVR rate after 24 weeks of treatment with IFN-α monotherapy is only 6%, and increases to only 13–19% after 48 weeks of treatment [8,9]. Ribavirin, an orally administered guanosine analog, is reported to have an excellent antiviral effect in chronic hepatitis C patients [10], and when combined with IFN-α2b for 48 weeks, it achieved an SVR rate of 38% [8]. Further antiviral advancement was achieved with the development of pegylated-IFN. Phase 3 trials evaluating the combination of pegIFN-α2a and ribavirin, and pegIFN-α2b and ribavirin revealed that they achieved SVR rates of 42–46% in patients infected with HCV genotype 1 and 76–82% in those infected with HCV genotype 2 or 3 [11,12]. In 2011, the first-generation orally bioavailable NS3-4A protease inhibitor, telaprevir, was approved for use in combination with pegIFN-α and ribavirin for patients infected with HCV genotype 1 [13]. Since 2014, the availability of IFN-free therapies combining multiple DAAs for patients with HCV-related chronic hepatitis or compensated cirrhosis has increased the SVR rate to over 95% [14,15]. In addition, a 12-week regimen of sofosbuvir plus velpatasvir was approved for patients with HCV-related decompensated cirrhosis in 2019 [16].

Since the approval of IFN-based therapy, numerous studies have reported the protective effect of anti-HCV therapy against hepatocarcinogenesis. For example, a study by Ikeda et al. involving 1643 chronic hepatitis C patients (1191 treated with IFN therapy and 452 without IFN therapy) found the rate of 10-year cumulative HCC incidence to be 12.0% in untreated patients, 15.0% in IFN therapy non-responders, 2.0% in incomplete responders who did not achieve SVR but achieved normal ALT levels, and 1.5% in patients who achieved SVR [17]. A prospective randomized controlled study on the effects of IFN therapy against hepatocarcinogenesis found that in patients with HCV-related compensated cirrhosis, IFN therapy significantly improved liver function and decreased the incidence of HCC (4% in IFN-treated patients versus 17% in untreated patients) [18]. These findings indicated that IFN-mediated eradication of HCV may contribute to the reduction of HCC development in patients with chronic HCV infection.

## 3. Suppression of HCC Development by DAA Therapy

It is well recognized that HCV eradication using IFN therapy can effectively reduce the risk of HCC development. However, a series of studies in 2016 reported an unexpectedly high rate of HCC incidence in patients with HCV-related chronic liver disease after SVR achievement using DAAs. Conti et al. reported that HCC developed in 26 of 344 (7.6%) cirrhotic patients who had been treated with DAA and followed for 24 weeks, suggesting that DAA-mediated eradication of HCV did not reduce HCC occurrence [19]. Similarly, DAA therapy seems to increase de novo HCC occurrence in patients with HCV-related cirrhosis [20]. These reports have raised concern that DAA therapy promotes HCC development after SVR. However, later studies involving large cohorts of HCV-positive patients and longer follow-up periods after SVR achievement suggest that DAA treatment does not increase the risk of HCC development and that, similar to IFN therapy, it may suppress HCC development [21]. Indeed, several studies clearly demonstrated that there was no difference in the rates of early HCC recurrence in patients who received IFN therapy relative to those who received DAA therapy, and that there were no significant differences between DAA regimens [22,23]. A retrospective cohort analysis of 17,836 patients revealed that both IFN therapy and DAA therapy significantly reduced the risk of hepatocarcinogenesis when compared with no treatment. Moreover, in cirrhosis patients who achieved SVR, those treated with either IFN therapy or DAA therapy had a significantly lower HCC incidence rate when compared with untreated patients [24].

Several recent reports show that IFN-free DAA therapy has the same inhibitory effect on hepatocarcinogenesis as IFN-based therapy and that it improves patient prognosis as much as IFN-based therapy. A large cohort study of 22,500 patients treated with DAAs (mean age 61.6 years, 39.0% with cirrhosis, 19,518 with SVR, and 2982 without SVR) indicated that the risk of hepatocarcinogenesis was reduced significantly in patients who achieved SVR (hazard ratio (HR): 0.28; 95% confidence interval (CI): 0.22–0.36) [25]. Another cohort study of 62,354 patients (35,871 treated with IFN, 4535 treated with IFN+DAA, and 21,948 treated with IFN-free DAA) showed that the inhibition of hepatocarcinogenesis upon achieving SVR with DAA therapy was as significant as when using IFN or IFN+DAA therapy [5].

## 4. Risk Factors for HCC Development after HCV Eradication Using DAA Therapy

Numerous studies have identified the clinical factors associated with the risk of HCC development or recurrence after DAA-mediated SVR [26].

Several studies have shown that liver cirrhosis is closely associated with the risk of HCC development and, consistently, that liver fibrotic markers are useful for predicting HCC after SVR achievement using DAA therapy. It is reported that before treatment of patients with cirrhosis with DAAs, those with pre-SVR FIB-4 scores of ≥ 3.25 had a higher annual HCC incidence than those with FIB-4 scores of < 3.25. In cases of advanced fibrosis, FIB-4 scores of > 3.7 at baseline and FIB-4 scores of > 3.3 one year after treatment were associated with de novo HCC [27]. A non-invasive method of measuring liver stiffness using ultrasonography revealed that those with a liver stiffness measurement (LSM) of ≥ 20 kPa at follow-up or those with LSM values of 10–20 kPa and albumin levels of < 4.4 g/dL were at the highest risk of HCC development after DAA treatment [28,29]. A recent evaluation of clinical outcomes following DAA therapy in patients with compensated and decompensated cirrhosis found that for patients with SVR, the cumulative HCC-free survival at 2 years for those with Child-Pugh A cirrhosis and Child-Pugh B/C cirrhosis was 94.5 and 87.6%, respectively [30].

A history of HCC treatment is another high risk factor for HCC development after DAA therapy. For example, a multicenter prospective study revealed that in patients who had received curative HCC treatment for a median duration of 17 months, de novo HCC occurred in 48 of 1161 (4.1%) patients, while HCC recurrence occurred in 40 of 124 (32%) patients [31]. Interestingly, HCC recurrence was significantly higher in patients with a history of more than two HCC treatments when compared with those with only one treatment [32]. Because of the high HCC recurrence rate even after HCV eradication, it is debatable whether DAA therapy offers SVR benefits to patients with HCC history. However, accumulating evidence suggests that DAA therapy may improve clinical outcomes in patients after curative HCC treatment. A study by Cabibbo et al. examined whether DAA therapy improved overall survival (OS) in patients with HCV-related cirrhosis after successful treatment for early-stage HCC and found that DAA therapy significantly improved OS and reduced the rate of hepatic decompensation. However, HCC recurrence was not significantly different between the DAA and no DAA groups [33]. A multicenter retrospective cohort study revealed that, compared with patients who did not receive antiviral therapy, mortality risk was significantly lower in patients with a history of HCC who achieved SVR after DAA therapy [34]. Dang et al. demonstrated that SVR was independently associated with a 60–70% risk reduction in both all-cause mortality and liver-related mortality in patients with HCV-related HCC [35]. Taken together, these findings indicate that DAA therapy could offer survival benefit even after curative HCC treatment, although the development of predictive markers of HCC recurrence is required.

Previous studies also identified various factors associated with the increased incidence of HCC after DAA-mediated SVR. Hepatic steatosis is known to increase the incidence of HCC, and it was shown that nonalcoholic fatty liver disease (NAFLD) is associated with increased incidence of HCC in chronic HCV patients after viral eradication [36,37,38]. Excessive drinking has been shown to significantly increase the prevalence of metabolic syndrome and is an important carcinogenic risk factor. In the multicenter cohort study with 2055 patients with HCV, 75 patients developed HCC during the mean observation period of 4.1 years after antiviral therapy, and obesity (BMI ≥ 25 kg/m^2^) and heavy alcohol consumption (≥60 g/day) were significant risk factors for development of HCC after SVR [39]. In addition, advanced age is a high risk factor for the development of HCC, and patients aged ≥75 years should continue careful surveillance for HCC even after HCV elimination [40]. From the viewpoint of the laboratory findings, patients with HCC development or recurrence after SVR were characterized by lower platelet counts (<80,000/μL), low albumin levels (<3.9 g/dL), and elevated AFP levels (>3.3 ng/mL) on post-SVR blood tests [41]. Debes et. al. evaluated serum levels of immune mediators before, during, and after DAA treatment for HCV infection. Comparing patients who developed HCC after DAA treatment with controls, 12 immune mediator (cytokines, growth factors, and apoptosis markers) levels were found to be significantly higher in serum before DAA treatment [42]. Although it was revealed that patients with genotype 3 tend to have a higher incidence of HCC [43], other papers reported that no significant difference was observed in HCV genotype as a virological factor associated with developing HCC after SVR [44,45].

## 5. Molecular Basis of Post-SVR Hepatocarcinogenesis: Genetic Alterations

The clinical observations described here raise the question of why HCC develops even after the eradication of HCV, a major putative HCC carcinogen. To answer this question, it is important to note that cancer development is based on the accumulation of genetic aberrations [46]. Recent international projects on comprehensive genetic analyses have elucidated the mutational landscape of liver cancers, including HCC [47,48,49,50,51]. One of the most common genetic alterations in HCC is *TERT* gene-associated aberrations, including promoter mutations and chromosomal translocations. Other major molecular factors altered in HCC include Wnt/β-catenin, p53/cell cycle, PI3K/Akt/mTOR, and RAS/RAF/MAPK signaling pathways, as well as chromatin-remodeling factors [47]. In addition, genes involved in various other processes, including oxidative stress, TGF-β signaling, and liver differentiation are also mutated in some HCC subtypes [52]. Since HCC tumorigenesis is considered to be driven by a stepwise accumulation of genetic aberrations in the chronically damaged liver, the genetic alterations accumulated in HCV-infected cirrhotic livers and early-stage HCC might provide clues about the genetic basis of hepatocarcinogenesis after HCV eradication [53].

We previously carried out whole exome sequencing analyses on non-cancerous cirrhotic liver tissues and found that HCV-positive cirrhotic livers frequently harbor numerous somatic mutations on various genes, including tumor-related genes [54]. To elucidate the landscape of genetic alterations in cirrhotic liver, we recently conducted a comprehensive genetic analysis of more than 200 regenerative nodules of cirrhotic liver tissues [55]. Targeted deep sequencing clearly showed that HCC-related somatic mutations are harbored by some of the tumor-related genes with low allelic frequencies, including *TP53*, *CTNNB1*, and *ARID1A*, although *TERT* promoter mutations, the most common genetic change in HCC, were not detected. These findings are consistent with those of two other large-scale genetic analyses on cirrhotic liver tissues [56,57]. On one hand, in a genetic analysis of non-tumor liver tissues of METAVIR stage F1 to F4, Zhu et al. identified significantly mutated genes in cirrhotic liver tissues, including *ARID1A* loss of function mutations. On the other hand, Brunner et al. conducted a whole genome sequencing (WGS) study on 482 microdissections from five normal and nine cirrhotic liver tissues and found that cirrhotic liver tissues not only had single nucleotide variations but also harbored structural chromosomal variations, including chromothripsis, a chromosomal crisis event associated with carcinogenesis.

The fact that various somatic mutations latently accumulate in the liver when chronically damaged by HCV infection prompted us to explore the mutational landscape of earlier-stage hepatocarcinogenesis. Using WGS, we evaluated the multistep accumulation of HCC-related genetic alterations in nodule-in-nodule HCC specimens consisting of progressed hypervascular HCC developed in the early hypovascular tumor arising from a common origin [58]. Notably, *TERT*-associated genetic changes were generally observed in the early stage of hepatocarcinogenesis, while case-specific cell-cycle/cell proliferation-associated pathways were altered in the progressive phase of multistage hepatocarcinogenesis. Importantly, early HCC tissues had already acquired numerous somatic mutations, including single nucleotide variations as well as dynamic chromosomal alterations such as copy number alterations, long deletions, chromosomal translocations, and even chromothripsis [58].

Taken together, these findings indicate that non-cancerous liver tissues, mainly liver cirrhosis due to chronic HCV infection, possess a somatic mutational burden and have already entered the multistep process towards hepatocarcinogenesis (Figure 1) [59]. Further analyses would be required to assess the comprehensive genetic aberrations in the liver tissue after HCV eradication.

## 6. Molecular Basis for Post-SVR Hepatocarcinogenesis: Epigenetic Alterations

It is also known that HCV infection can induce epigenetic alterations to hepatocytes underlying hepatitis and/or cirrhosis [60,61,62,63]. Interestingly, Hamdane et al. reported that HCV-induced epigenetic changes with liver cancer risk persist after SVR [64]. Through ChIP sequencing of post-SVR and HCV-positive livers, they showed that HCV-induced modifications of the histone mark, H3K27ac, persist in human liver samples after DAA-mediated HCV cure. Integrated analysis of histone modification and gene expression data revealed that *SPHK1* upregulation remains after HCV eradication, and interestingly, the high expression of *SPHK1* is significantly associated with HCC risk after SVR. To elucidate the landscape of transcriptional changes in post-SVR livers, we recently conducted total transcriptomic analysis on post-SVR livers [65]. Comparison of total gene expression data in post-SVR livers relative to HCV-positive and normal liver tissues revealed that some oncogenic pathways are upregulated in post-SVR liver tissues. Interestingly, some abnormal gene expression profiles caused by HCV infection did not return to normal even after HCV eradication. Consistently, in vitro experiments using HuH7, a human liver cell line that was infected once with the JFH-1 strain of HCV and then treated with DAA to eradicate HCV, revealed that various oncogenic pathways were upregulated upon JFH-1 infection and that some pathways remained upregulated even after complete HCV eradication. The oncogenic pathways that were sustained after SVR are associated with cell proliferation, cell adhesion, the cell cycle, and inflammation (Figure 2). Taken together, these findings indicate that genetic, epigenetic, and transcriptional alterations caused by chronic HCV infection are maintained even after viral eradication, and that aberrations with malignant potential might remain imprinted in the liver even after HCV eradication. Interestingly, a previous study revealed that HCC tumorigenicity can stem from a metabolic plasticity, allowing them to thrive in a broader range of glucose concentrations [66], suggesting that metabolomics and proteomics in addition to transcriptomics on the liver tissues after SVR are necessary for further analyses. Although genetic, epigenetic, and transcriptional alterations caused by HCV infection are a potential molecular basis for post-SVR hepatocarcinogenesis, there are only a few multi-omics studies on post-SVR cases. However, because each of these studies involved small patient cohorts, omics-based studies with larger cohorts are needed to validate the findings from multi-omics profiles associated with post-SVR carcinogenesis and to elucidate the molecular basis of post-SVR HCC. In addition, because HCV infection and SVR are tightly associated with immunologic responses to the virus, new techniques, such as single-cell RNA sequencing and spatial gene expression mapping technologies, are powerful tools for elucidating the pathogenesis of post-SVR liver tissues and the molecular basis of hepatocarcinogenesis after HCV eradication [67,68].

## 7. Conclusions

Because chronic inflammation is a major cause of tumorigenesis, the elimination of pathogens that induce chronic inflammation reduces the incidence of tumors, although the risk of tumorigenesis persists. For example, *Helicobacter pylori* (*H. pylori*) is a well-known risk factor for gastric cancer, and although its eradication reduces gastric cancer incidence, 0.35% of patients who achieve *H. pylori* eradication develop gastric cancer [69]. We previously found that genomic aberrations accumulate in gastric mucosa infected with *H. pylori*, suggesting that non-tumor gastric tissue with chronic inflammation is highly susceptible to carcinogenesis [70]. Therefore, persistent inflammation driven by chronic infection, including by viruses and bacteria, may cause the accumulation of genetic or epigenetic aberrations in various organs, leading to tumorigenesis even after the pathogens are eradicated. Because HCC may develop in patients who achieve HCV eradication, periodic HCC screening is necessary after HCV eradication, especially in cases with liver cirrhosis and/or history of HCC treatment.

## Figures and Tables

**Figure 1 pathogens-11-00430-f001:**
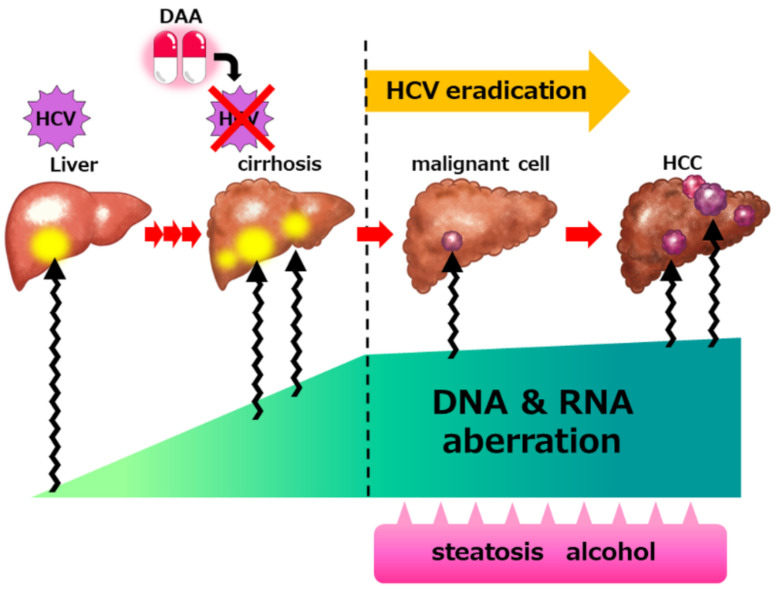
Accumulated genetic alterations in hepatitis C virus (HCV)-infected liver contributes to hepatocarcinogenesis even after HCV eradication. Persistent HCV infection and associated chronic inflammation results in the stepwise accumulation of genetic and epigenetic aberrations. Although achieving sustained viral response (SVR) with direct-acting antiviral (DAA) treatment suppresses the progression of liver fibrosis and carcinogenesis, accumulated genetic aberrations may lead to hepatocellular carcinoma (HCC) development after HCV eradication. In addition, liver fat accumulation, impaired glucose tolerance, and alcohol consumption may accelerate carcinogenesis.

**Figure 2 pathogens-11-00430-f002:**
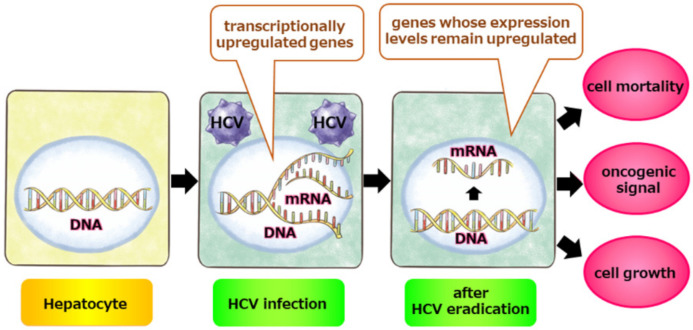
Irreversible transcription profiles even after HCV eradication. A subset of transcriptional changes caused by HCV infection can remain in hepatocytes even after HCV eradication. Such transcriptional alterations may contribute to enhanced cell proliferation, promotion of apoptosis, and dysregulation of intracellular signaling.

## Data Availability

No new data were created or analyzed in this study. Data sharing is not applicable to this article.

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
