# Peer review of "Clinical and Molecular Basis of Hepatocellular Carcinoma after Hepatitis C Virus Eradication"

_pathogens, 2022, doi:10.3390/pathogens11040430_

Round 1

Reviewer 1 Report

This review by Oe et al. is clear, well written and very interesting to the field of HCC, and most importantly bring new valuable insights. Here is my comment:

  • In this paragraph, authors said: “Consistently, in vitro experiments using HuH7, a human liver cell line that was infected once with the JFH-1 strain of HCV and then treated with DAA to eradicate HCV, revealed that various oncogenic pathways were upregulated upon JFH-1 infection and that some pathways remained upregulated even after complete HCV eradication. The oncogenic pathways that were sustained after SVR are associated with cell proliferation, cell adhesion, the cell cycle, and inflammation.” Please add more details, for example what about metabolism of these cells? Did glucose metabolism is altered due to fibrosis? What about the amount of ATP and OXPHOS? I do think that a paragraph dealing with the metabolism of these cells is lacking and thus authors should discuss a little more about it. To do so, please refer and cite this following paper which clearly demonstrated that HCC cell metabolism/metabolic rewiring is important for their increased tumorigenicity: Cassim S, Raymond VA, Dehbidi-Assadzadeh L, Lapierre P, Bilodeau M. Metabolic reprogramming enables hepatocarcinoma cells to efficiently adapt and survive to a nutrient-restricted microenvironment. Cell Cycle. 2018;17(7):903-916. doi: 10.1080/15384101.2018.1460023. Epub 2018 May 21. PMID: 29633904; PMCID: PMC6056217.

Author Response

We thank the reviewer for his/her thoughtful comments, which have allowed us to improve the paper. We have addressed the reviewer’s criticisms through the following changes to the manuscript.

  1. Please add more details, for example what about metabolism of these cells? Did glucose metabolism is altered due to fibrosis? What about the amount of ATP and OXPHOS? I do think that a paragraph dealing with the metabolism of these cells is lacking and thus authors should discuss a little more about it. To do so, please refer and cite this following paper which clearly demonstrated that HCC cell metabolism/metabolic rewiring is important for their increased tumorigenicity. Metabolic reprogramming enables hepatocarcinoma cells to efficiently adapt and survive to a nutrient-restricted microenvironment. (Cell Cycle. 2018;17(7):903-916).

Reply:

According to the reviewer’s suggestion, we cited the recommended paper (reference#66) as follows in the revised manuscript (page6, lines 241-245):

“Interestingly, previous study revealed that HCC tumorigenicity can stem from a metabolic plasticity allowing them to thrive in a broader range of glucose concentrations (66), suggesting that metabolomics and proteomics in addition to transcriptomics on the liver tissues after SVR are necessary.”

Reviewer 2 Report

This concise review discusses the clinical and molecular basis of HCC after HCV cure by DAA therapy. The review is well written and includes most recent data relevant to this topic. However, it is a little confusing as to why only a subset of patients continue to develop HCC if HCV causes long-lasting genetic and epigenetic changes in liver cells. Also, the authors pointed out several risk factors and discussed the effect of HCC history on HCC development after HCV eradication in greater detail. However, it would be good to also discuss in more detail regarding other risk factors (advanced age, male sex, elevate total bilirubin levels, low albumin level, low platelet count, elevated AFP level). In addition, is HCV genotype a risk factor? Lastly, the sentence on page 3, line 120-122 is confusing and needs to be modified.

Author Response

We thank the reviewer for his/her thoughtful comments, which have allowed us to improve the paper. We have addressed the reviewer’s criticisms through the following changes to the manuscript.

  1. It is a little confusing as to why only a subset of patients continue to develop HCC if HCV causes long-lasting genetic and epigenetic changes in liver cells.

Reply: 

There are several possible causes for the individual differences in the timing and recurrence rate of HCC due to HCV infection.

First, the period between HCV infection and HCV elimination varies from patient to patient. Second, the severity of hepatic inflammation caused be HCV infection also varies from patient to patient.  Some cases have little or no chronic inflammation (persistent normal alanine transaminase) and thus accumulate genetic abnormalities at a slower pace than those with persistent chronic inflammation. Furthermore, the presence or absence of carcinogenic risks other than HCV infection, such as alcohol consumption or steatohepatitis, could contribute the difference in the timing of HCC occurrence.

  1. Authors pointed out several risk factors and discussed the effect of HCC history on HCC development after HCV eradication in greater detail. However, it would be good to also discuss in more detail regarding other risk factors (advanced age, male sex, elevate total bilirubin levels, low albumin level, low platelet count, elevated AFP level). Is HCV genotype a risk factor?

Reply: 

According to the reviewer’s suggestion, we added the discussion in detail about other risk factors for HCC after SVR in the revised manuscript (page3, lines 139-160).

  1. The sentence on page 3, line 120-122 is confusing and needs to be modified.

Reply: According to the reviewer’s suggestion, we corrected the confusing description of the revised manuscript as follows (page3, lines 121-122):

“For example, a multicenter prospective study revealed that in patients who had received curative HCC treatment for a median duration of 17 months, de novo HCC occurred in 48 of 1,161 (4.1%) patients while HCC recurrence occurred in 40 of 124 (32%) patients.”

Reviewer 3 Report

Clinical and molecular basis of hepatocellular carcinoma 2 after hepatitis C virus eradication

In this review, the authors summarized the understanding of chronic hepatitis C virus treatment, the cure rate of the virus, and the risk of hepatitis C virus-related HCC development. This review outlines the most recent data about different predictive factors for HCC development after treatment of hepatitis C virus-infected patients with a specific focus on molecular factors associated with post-SVR hepatocarcinogenesis (genetic and epigenetic alterations). The manuscript is clearly and well written. In general, I found this review very interesting.

I would recommend only the following minor changes:

  • The summary of risk/predictive factors for the development of HCC after HCV eradication is missing several important factors as are immunological markers or the factors related to underlying metabolic-associated fatty liver diseases. Therefore, I would recommend completing the sentence (line 102-105) “Numerous studies have identified the clinical factors associated with the risk of HCC development or recurrence after DAA-mediated SVR, including advanced age, male sex, elevated total bilirubin levels, low albumin level, low platelet count, elevated AFP level, history of HCC (26) and other factors as recently reviewed elsewhere https://doi.org/10.3390/livers1040024 ”
  • The conclusions on lines 137-138 “Taken together, these findings indicate that DAA therapy offers survival benefit after curative HCC treatment.” are surprising as the conclusions of the chapter called “Risk factors for HCC development after HCV eradication using DAA therapy” I would recommend rather finishing by mentioning the need of integrated predictive models combining different markers: predictive factors.
  • Recent key study related to this review can be also included: https://doi.org/10.1111/jgh.15243
  • What do the authors mean by the statement „HCV infection can induce epigenetic alterations to hepatitis“? This sentence maybe needs to be reformulated.

Author Response

We thank the reviewer for his/her thoughtful comments, which have allowed us to improve the paper. We have addressed the reviewer’s criticisms through the following changes to the manuscript.

  1. The summary of risk/predictive factors for the development of HCC after HCV eradication is missing several important factors as are immunological markers or the factors related to underlying metabolic-associated fatty liver diseases. Therefore, I would recommend completing the sentence (lines 102-105) “Numerous studies have identified the clinical factors associated with the risk of HCC development or recurrence after DAA-mediated SVR, including advanced age, male sex, elevated total bilirubin levels, low albumin level, low platelet count, elevated AFP level, history of HCC (26) and other factors as recently reviewed elsewhere. Predictive Factors for Hepatocellular Carcinoma Development after Direct-Acting Antiviral Treatment of HCV. (Livers 2021: 1(4); 313-321.)

Reply: 

According to the reviewer’s suggestion, we added the discussion in detail about other risk factors for HCC after SVR (page3, line 139-160) and also added the suggested review paper in the revised manuscript (reference#26).

  1. The conclusions on lines 137-138 “Taken together, these findings indicate that DAA therapy offers survival benefit after curative HCC treatment.” are surprising as the conclusions of the chapter called “Risk factors for HCC development after HCV eradication using DAA therapy” I would recommend rather finishing by mentioning the need of integrated predictive models combining different markers: predictive factors.

Reply: 

According to the reviewer’s comments, we modified the sentence as follows (page3, lines 136-138):

“Taken together, these findings indicate that DAA therapy could offer survival benefit even after curative HCC treatment, although the development of integrated predictive models combining different markers for HCC recurrence is required.”

  1. Recent key study related to this review can be also included. Impact of liver-stiffness measurement on hepatocellular carcinoma development in chronic hepatitis C patients treated with direct-acting antivirals: A systematic review and time-to-event meta-analysis. (J Gastroenterol Hepatol. 2021: 36(3); 601-608.)

Reply:

According to the reviewer’s suggestion, we added the recommended paper (reference#29).

  1. What do the authors mean by the statement „HCV infection can induce epigenetic alterations to hepatitis“? This sentence maybe needs to be reformulated.

Reply:

According to the reviewer’s suggestion, we modified the sentence in the revised manuscript as follows (page5, lines 220-221.);

“It is also known that HCV infection can induce epigenetic alterations to hepatocytes underlying hepatitis and/or cirrhosis. “